

# Evaluating the readability, quality and reliability of online patient education materials on post-covid pain

Erkan Ozduran[1] and Sibel Büyükçoban[2]

[1] Department of Physical Medicine and Rehabilitation, Algology, Dokuz Eylül University, Izmir, Turkey
[2] Department of Anesthesiology and Reanimation, Dokuz Eylül University, Izmir, Turkey

Corresponding authors
Erkan Ozduran,
erkanozduran@gmail.com
Sibel Büyükçoban,
sbuyukcoban101@gmail.com

## ABSTRACT

**Background**. The use of the Internet to access healthcare-related information is increasing day by day. However, there are concerns regarding the reliability and comprehensibility of this information. This study aimed to investigate the readability, reliability, and quality of Internet-based patient educational materials (PEM) related to "post-COVID-19 pain."

**Methods**. One-hundred websites that fit the purposes of the study were identified by searching for the terms "post-COVID-19 pain" and "pain after COVID-19" using the Google search engine on February 24, 2022. The website readability was assessed using the Flesch Reading Ease Score (FRES), Flesch–Kincaid Grade Level (FKGL), Simple Measure of Gobbledygook (SMOG), and Gunning FOG (GFOG). The reliability, quality, and popularity of the websites were assessed using the JAMA score, DISCERN score/Health on the Net Foundation code of conduct, and Alexa, respectively.

**Results**. Upon investigation of the textual contents, the mean FRES was $51.40 \pm 10.65$ (difficult), the mean FKGL and SMOG were $10.93 \pm 2.17$ and $9.83 \pm 1.66$ years, respectively, and the mean GFOG was $13.14 \pm 2.16$ (very difficult). Furthermore, 24.5% of the websites were highly reliable according to JAMA scores, 8% were of high quality according to GQS values, and 10% were HONcode-compliant. There was a statistically significant difference between the website types and reliability ($p = 0.003$) and quality scores ($p = 0.002$).

**Conclusion**. The readability level of PEM on post-COVID-19 pain was considerably higher than grade 6 educational level, as recommended by the National Institutes of Health, and had low reliability and poor quality. We suggest that Internet-based PEM should have a certain degree of readability that is in accordance with the educational level of the general public and feature reliable content.

## INTRODUCTION

Coronavirus disease 2019 (COVID-19), which is caused by the severe acute respiratory syndrome coronavirus 2 (SARS-CoV-2) pathogen, led to a worldwide medical and humanitarian crisis. In November 2019, the first case was recorded in Wuhan, China, and WHO reported the first case on 31 December 2019. In March 2021, the epidemic

was declared a global pandemic, and by 30 May 2020, 899,866 positive cases and 364,891 deaths were detected (*Adil et al., 2021*). The symptoms associated with COVID-19 have not only been observed in the respiratory system but also in the muscular, neurological, and cardiovascular systems. In a Chinese study, the associated symptoms included fever (88.7%), cough (67.8%), and fatigue (38.1%), in order of prevalence (*Yang et al., 2020*). The World Health Organization declared that approximately 15% of the patients experienced myalgia and arthralgia in the scope of the symptoms associated with COVID-19 (*Fernández-de Las-Peñas et al., 2021*). It was also suggested that myalgia and arthralgia were the fifth most prevalent symptoms in the acute period of COVID-19 and may become chronic (*Struyf et al., 2020*).

Despite the fact that COVID-19 was initially considered a short-term disease, it was subsequently revealed that many post-treatment symptoms persisted with manifestations called post-COVID-19 or long COVID (*Nabavi, 2020*). The term "prolonged COVID-19" has been used for cases in which the patient survived COVID-19 but had persistent effects of infection or experienced symptoms lasting for more than 1 month (*Baig, 2021*). In the case of long COVID-19, it has been reported that the most common symptom is fatigue with 73%, joint or muscle pain is in the 4th place with 49%, and headache is in the 6th place with 33% (*Jacques et al., 2022*). *Şahin et al. (2021)* stated that the complaints of pain after COVID-19 disease continued until the 11th week. In these cases where the complaints are prolonged, there is no consensus in terms of diagnosis and management (*Greenhalgh et al., 2020*). It is evident that the correct treatment algorithm would help individuals with recovery given that pain symptoms adversely affect the quality of life during the post-COVID-19 period.

Patients can rapidly access the desired healthcare content using Internet-based patient educational materials (PEM), which have recently been used as an important tool for acquiring further information (*Agar & Sahin, 2021*; *Guo et al., 2019*). In 2018, it was reported that 90% of the adults in the United States used the Internet and that three-quarters performed healthcare-related searches (*Guo et al., 2019*). The National Institutes of Health, the US Department of Health and Human Services, and the American Medical Association reported that Internet-based PEM should be developed below the sixth-grade educational level (*Guo et al., 2019*; *Wang, Capo & Orillaza, 2009*). If readability of online information posted on a website is above the said grade, it may be considered difficult to read and understand for an average reader. Therefore, it is important that the healthcare-related information on the websites are compliant with the average educational level of the readers and carefully evaluated before release. Access to online information increases daily, but this raises concerns about the accuracy, reliability, and quality of the said information and whether an appropriate level of readability is offered. Relevant studies in the literature investigated the quality and readability of the information included in Internet-based PEMs on a number of medical conditions (*Han & Carayannopoulos, 2020*; *Basch et al., 2020*). A study by *Worrall et al. (2020)* reported that the readability level of online information about COVID-19 was poor and difficult to read. Only 17.2% ($n = 165$) of all analyzed readability scores showed a universally readable level. Average readability scores of searched web

pages from all regions (Ireland, United Kingdom, United States and Canada) were below standard universal readability levels (*Worrall et al., 2020*).

It is well established that patients furnished with information about the etiology, pathophysiology, treatment, and prevention methods would more likely participate in and comply with the disease prevention or treatment procedures (*Ahmed et al., 2020*). It is evident that providing individuals with reliable, high-quality, and readable online information about post-COVID-19 pain would help with the management of a condition that affects many people (*Basch et al., 2020*; *Şahin et al., 2021*; *Jacques et al., 2022*). This study aimed to investigate websites containing PEM on post-COVID-19 pain based on their readability, quality, and reliability. Furthermore, it aimed to investigate the website types that provided highly reliable information on post-COVID-19 pain.

## MATERIALS & METHODS

This Study was planned as a cross-sectional study. On February 24, 2022, the terms "pain after COVID-19," "post-COVID-19 pain," and "long COVID-19 and pain" were searched by two authors (E.O. and S.B.) using the Google search engine (www.google.com), which is the most popular search engine. A collective assessment (Joint evaluation between authors) was used to reach a final decision in case of any inconsistency between the authors during the assessment of the websites. Google search engine was used because based on data from December 2021, Google led the search engine sector with a market share of 86.19% (*Johnson, 2022*).

### Websites selection criteria

The cookies were removed and the computer's browser history was deleted during the website search to ensure that the search results were not affected (such as by Google Ads). To avoid bias based on search history, searched region, and cookies, access was done with Google Chrome's incognito form. In addition, the searches were made after signing off from all Google accounts. Following the each search, the uniform resource locators (URLs) of the first 100 websites that met the inclusion criteria were recorded, consistent with the methodologies of similar studies in the relevant literature (*Basch et al., 2020*; *Jayasinghe et al., 2020*). The 10 websites that appeared on the first page of the search results were considered the most viewed websites (*Eysenbach & Köhler, 2002*). Websites with non-English language content, those without information on post-COVID-19 pain, those that required registration or subscription, repetitive websites, those with video or audio recording content but without text content, and journal articles were not included in the study. Furthermore, the graphics, images, videos, tables, figures, and list formats contained in the text, all punctuation marks, URL websites, author information, references to avoid erroneous results, addresses, and phone numbers were not included in the assessment (*Zeldman, 2001*). From the 195 websites we obtained after these search terms, 54 duplicates were removed and 151 websites remained. According to the exclusion criteria, 41 more websites were removed and 100 websites were included in our study (Fig. 1).

In case there was no evaluation criterion on the home page, the three-click rule was applied during the assessment of the websites (*Charnock et al., 1999*). This rule states that

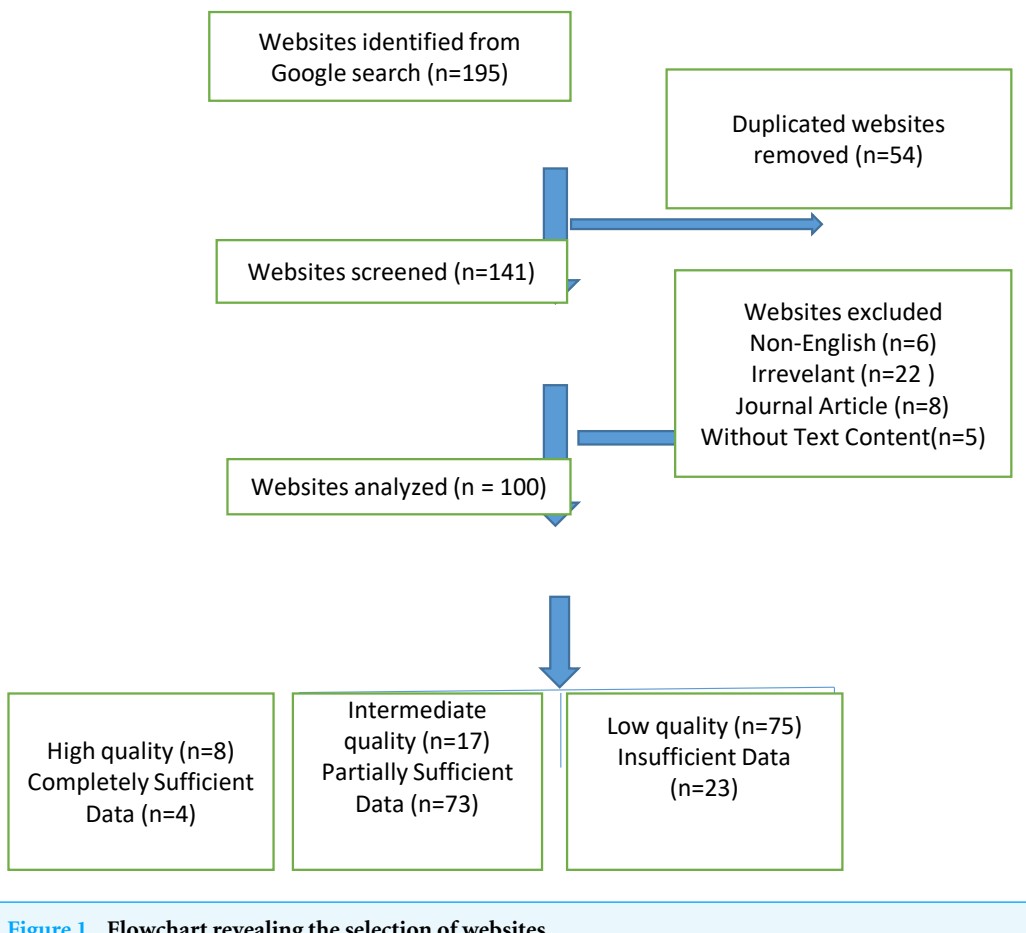

**Figure 1** **Flowchart revealing the selection of websites.**

the website user would find any information in up to three mouse clicks. Although it is not an official rule, it is considered that if the information cannot be accessed by three clicks, the user cannot reach one's goal and would leave the website.

## Ethical considerations

Our study was conducted with the approval of the Non-Interventional Research Ethics Committee (6958-GOA 2022/06-09).

## Website typology

Based on their type and ownership, the websites were classified into six categories by the two authors (*Guo et al., 2019*). In addition, depending on the URL extension (com., net, gov., edu., org.) websites were tried to be determined whether they are from professional institutions, government, commercial *etc* (*Basch et al., 2020*). Typologies were professional (websites created by organizations or individuals with professional medical qualifications, URL extension-edu, com), commercial (websites that sell product for profit (URL extension-com, net), nonprofit (non-profit educational/charitable/supporting sites, URL extension-org), health portals (websites that provide information about health issues, URL extension-com, net), news (news and information created to provide magazine

websites or newspaper URL extension-com, net), government (websites created, regulated or administered by an official government agency, URL extension-gov) (*Yurdakul, Kilicoglu & Bagcier, 2021*; *Daraz et al., 2018*).

## Journal of American Medical Association (JAMA) benchmark criteria

The JAMA benchmarks analyzes online information and resources under 4 criteria: authorship, attribution, disclosure, and currency. The scorer awards 1 point for each criterion in the text, and the final score ranges from 0 to 4. Four points represent the highest reliability and quality (*Silberg, Lundberg & Musacchio, 1997*). A website with a JAMA score of ≥3 points was considered highly reliable, whereas those with a JAMA score of ≤2 points were considered to have low reliability (*Silberg, Lundberg & Musacchio, 1997*) (Table 1).

## DISCERN criteria

The DISCERN criteria, a tool used to indicate the quality of websites, consist of 16 items that are scored between 1 and 5 (*Weil et al., 2014*). The two authors independently reviewed the websites based on the DISCERN criteria. The final DISCERN score for each website was reached after the scores by the two authors were averaged. The final DISCERN score ranged from 16 to 80. Based on the results, scores of 63 to 80 were considered excellent; 51 to 62, good; 39 to 50, fair; 28 to 38, poor; and 16 to 27, very poor (*Boyer, Selby & Appel, 1998*) (Table 1).

## Global quality score (GQS)

The quality of the websites was rated based on the GQS criteria, which makes use of a 5-point scale to assess the overall quality of a website. The scores refer to the informative quality of the website and to what extent the reviewer considers it useful for the patients. Accordingly, 1 point indicates poor quality and 5 points indicate excellent quality (*Agar & Sahin, 2021*) (Table 1).

## Health on the net foundation code of conduct (HONcode) certification

Established with an aim to promote the online distribution and efficient use of reliable and useful health information, The Health on the Net Foundation (HON) designed the HONcode to help standardize the reliability of healthcare-related information available on the Internet (*Boyer, Baujard & Geissbuhler, 2011*). To meet the HONcode criteria, the date and source of the content should be disclosed, the competencies of the authors should be specified, a privacy policy should be included, the website should complement the patient–physician relationship, the finances and advertising policy of the website should be transparent, and contact information should be provided (*Walsh & Volsko, 2008*). This study investigated whether there was a HONcode stamp posted on the main page or included in the related URL.

## Readability

The Flesch Reading Ease Score (FRES), Flesch–Kincaid grade level (FKGL), Simple Measure of Gobbledygook (SMOG), Gunning FOG (GFOG), Coleman–Liau (CL) score, automated readability index (ARI), and Linear Write (LW) readability formulas as retrieved from

**Table 1** Contents of JAMA, DISCERN and GQS assessment criteria.

| JAMA Benchmark Criteria | Total Score (0–4 Points) |
| --- | --- |
| Authorship | 1 point (Authors and contributors, their affiliations,and relevant credentials should be provided) |
| Attribution | 1 point ( References and sources for all content should be listed) |
| Disclosure | 1 point (Conflicts of interest, funding,sponsorship, advertising, support, and video ownership should be fully disclosed) |
| Currency | 1 point (Dates that on which the content was posted and updated should be indicated). JAMA is used to evaluate the accuracy and reliability of information) |
| **DISCERN Criteria** | **Total Score (16–80 Points)** |
| 1 Are the aims clear? | 1–5 point |
| 2 Does it achieve its aims? | 1–5 point |
| 3 Is it relevant? | 1–5 point |
| 4 Is it clear what sources of information were used | 1–5 point |
| 5 Is it clear when the information used or reported in the publication was produced? | 1–5 point |
| 6 Is it balanced and unbiased? | 1–5 point |
| 7 Does it provide details of additional sources of 1.45 support and information? | 1–5 point |
| 8 Does it refer to areas of uncertainty? | 1–5 point |
| 9 Does it describe how each treatment works? | 1–5 point |
| 10 Does it describe the benefits of each treatment? | 1–5 point |
| 11 Does it describe the risks of each treatment? | 1–5 point |
| 12 Does it describe what would happen if no treatment is used? | 1–5 point |
| 13 Does it describe how the treatment choices affect overall quality of life? | 1–5 point |
| 14 Is it clear that there may be more than one possible treatment choice? | 1–5 point |
| 15 Does it provide support for shared decision making? | 1–5 point |
| 16 Based on the answers to all of the above questions, rate the overall quality of the publication as a source of information about treatment choices. | 1–5 point |
| **GQS** | **Score** |
| Poor quality, poor flow of the site, most information missing, not at all useful for patients | 1 |
| Generally poor quality and poor flow, some information listed but many important topics missing, of very limited use to patients | 2 |
| Moderate quality, suboptimal flow, some important information is adequately discussed but others poorly discussed, somewhat useful for patients | 3 |
| Good quality and generally good flow, most of the relevant information is listed, but some topics not covered, useful for patients | 4 |
| Excellent quality and excellent flow, very useful for patients | 5 |

**Notes.**
JAMA, Journal of American Medical Association; GQS, Global Quality Score.

http://www.readability-score.com were used for the purpose of assessing the readability of websites (*Basch et al., 2020*; *Jayasinghe et al., 2020*) (Table 2). The results of FRES should correlate inversely with FKGL, SMOG, GFOG, CL, ARI and LW so that text with a high FRES has lower FKGL, SMOG, and GFOG scores. The acceptable readability level was set as $\geq 60.0$ for the FRE and $<7$ for the FKGL, SMOG, GFOG, CL, ARI and LW (*Yeung et al., 2022*).

The readability formulas were used in the assessment of all the text contents, except for the aforementioned exclusions (non-English contents, without information on post-COVID-19 pain, those with video or audio recording content but without text content, and journal articles). The ranking values of all the websites were rated and recorded. The texts were saved in Microsoft Office Word 2007 (Microsoft Corporation, Redmond, WA, USA). The average readability level based on all the readability formulas was compared based on the sixth-grade educational level as recommended by the American Medical Association and the National Institutes of Health.

### Popularity and visibility analysis

Alexa (https://www.alexa.com/) is a website popularity ranking that is often used to assess the visibility and popularity of the website (*Wald, Dube & Anthony, 2007*). It measures how often a website was clicked and visited during the past 3 months compared with other websites. Higher scores indicate higher popularity based on higher click rates.

### Content analyses

The websites were investigated and assessed by type based on whether a given website contained certain topics related to post-COVID-19 pain, such as etiology, diagnosis, non-pain-related symptoms, treatment, exercise, prevention, risk factors, and vaccine–pain relationship (*Han & Carayannopoulos, 2020*).

### Statistical analysis

For statistical analysis, data were uploaded to SPSS Windows 25.0 software (SPSS Inc., Chicago, IL). In our study dependent variables are readability scores, JAMA, DISCERN, GQS results, HONcode presence, ALEXA values and contents. The independent variables are "top 10 and remaining website grouping" and website typologies. Continuous values are indicated as mean $\pm$ SD, while frequency variables are given as number (n) and percentage (%). For statistical analysis, the Mann–Whitney U or Kruskal Wallis tests were used to compare groups with continuous values such as readability indices and sixth class level. For comparison of frequency variables, the Chi-square or Fisher exact tests were used. A $p$ value lower than 0.05 was accepted as statistically significant difference.

## RESULTS

### Website typologies

Upon comparing 100 websites that met the study's inclusion criteria by type, news (31%) and professional (29%) types were found to be the most common website types (Fig. 2). Previous studies reported that users were particularly interested in the results that appeared on the first page of a search engine. Google provides 10 search results on the first page.

Ozduran and Büyükçoban (2022), *PeerJ*, DOI 10.7717/peerj.13686

**Table 2  Readability indices and features.**

| Readability Index | Description | Formula |
|---|---|---|
| Flesch Reading Ease Score(FRES) | It was developed to evaluate the readability of newspapers. It is best suited for evaluating school textbooks and technical manuals. The standardized test used by many US government agencies. Scores range from 0 to 100, with higher scores indicating easier readability | $I = (206.835 - (84.6 \times (B/W)) - (1.015 \times (W/S)))$ |
| Flesch–Kincaid grade level (FKGL) | Part of the Kincaid Navy Personnel test collection. Designed for technical documentation and suitable for a wide range of disciplines | $G = (11.8 \times (B/W)) + (0.39 \times (W/S)) - 15.59$ |
| Simple Measure of Gobbledygook (SMOG) | It is generally suitable for middle-aged (4th grade to college level) readers. While testing 100% comprehension, most formulas test about 50%–75% comprehension. Most accurate when applied to documents $\geq$30 sentences long. | $G = 1.0430 \times \sqrt{C} + 3.1291$ |
| Gunning FOG (GFOG) | It was developed to help American businesses improve the readability of their writing. Applicable to many disciplines | $G = 0.4 \times (W/S + ((C*/W)X100))$ |
| Coleman–Liau (CL) score | It is designed for middle-aged (4th grade to college level) readers. The formula is based on text in the grade level range of 0.4 to 16.3. It applies to many industries. | $G = (-27.4004 \times (E/100)) + 23.06395$ |
| Automated readability index (ARI) | ARI has been used by the military in writing technical manuals, and its calculation returns a grade level necessary for understanding. | $ARI = 4.71 \times l + 0.5 * ASL - 21.43$ |
| Linsear Write (LW) | It is developed for the United States Air Force to help them calculate the readability of their technical manuals | $LW = (R + 3C)/S$<br>Result<br>• If >20, divide by 2<br>• If $\leq$20, subtract 2, and then divide by 2 |

**Notes.**

G, Grade level; B, Number of syllables; W, Number of words; S, Number of sentences; I, Flesch Index Score; SMOG, Simple Measure of Gobbledygook; C, Complex words ($\geq$3 syllables); E, predicted Cloze percentage=141.8401 - (0.214590 ×number of characters) + (1.079812 *S); C *, Complex words with exceptions including, proper nouns, words made three syllables by addition of "ed" or "es", compound words made of simpler words; ASL, the average number of sentences per 100 words; R, the number of words $\leq$2 syllables.

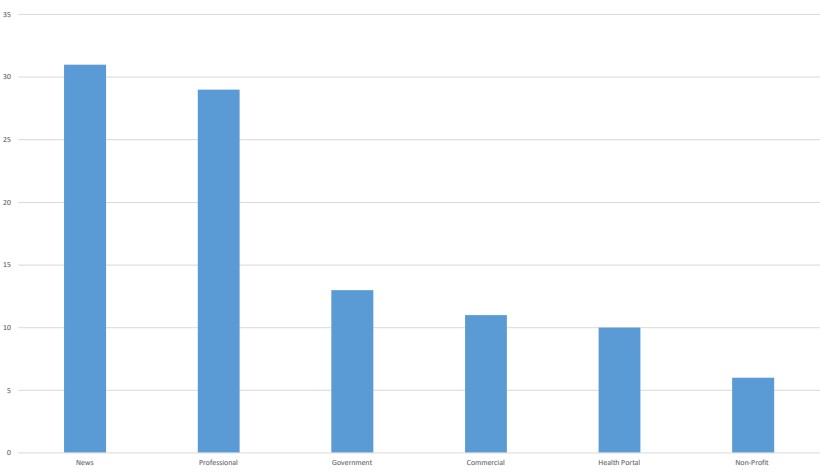

**Figure 2** Types of websites in the whole search.

There was a statistically significant difference in website types between the first 10 search results and the rest ($p = 0.043$). The fact that 60% of the first 10 websites were created by professional associations and institutions, whereas 31.4% of the remaining 90 websites were created by news websites, might account for the significant difference.

## Comparison of readability, reliability and quality scores of top 10 and other websites

There was no significant difference in readability between the top 10 and the remaining websites (FRES, GFOG, GFOG, CL, SMOG; $p > 0.050$). There was also no significant difference between the top 10 and the remaining websites in JAMA reliability ($p = 0.350$), DISCERN quality ($p = 0.613$), and HONcode compliance ($p = 0.267$) (Table 3). Nevertheless, there was a significant relationship between the top 10 and the remaining websites by GQS results ($p < 0.001$) (Table 3).

## Reliability and quality evaluation

The mean JAMA score, DISCERN score and GQS score of the 100 websites was $2 \pm 0.76$, $36.40 \pm 14.70$ and $2.18 \pm 0.85$, respectively. The results suggested that the websites had low reliability and poor quality. There was a significant relationship between the JAMA reliability scores ($p < 0.001$) and GQS values ($p < 0.001$) by website type among the 100 websites. This difference can be explained by the higher JAMA reliability scores of the websites created by nonprofit organizations and higher GQS values of the health portal-based websites. These scores were lower for the websites created by news channels. Further, 20% of the websites were rated as highly reliable based on a JAMA score of $\geq 3$ and 8% were identified as being high quality based on GQS values. HONcode compliance was noted in only 10% of the websites. The highest rate of HONcode compliance was noted in the health portals (7%) (Table 4). No significant difference was found between DISCERN scores and website typologies ($p = 0.207$). According to the website typologies, the reliability (JAMA) ranking is as follows, from the highest to the lowest; Non-Profit

**Table 3** Comparison of JAMA, DISCERN scores, HONcode presences and reading levels according to the typologies of the websites.

| | Professional | Commercial | Non-Profit | Health Portal | News | Government | p |
|---|---|---|---|---|---|---|---|
| **N(%)** | 29(29%) | 11(11%) | 6(6%) | 10(10%) | 31(31%) | 13(13%) | |
| **JAMA (Mean ± SD)** | 2.03 ± 0.73 | 1.63 ± 0.5 | 2.5 ± 1.22 | 2.4 ± 0.84 | 2.06 ± 0.57 | 1.53 ± 0.87 | **0.049** |
| Insufficient Data n:23 | 7(24.1%) | 4(36.4%) | 1(16.7%) | 1(10%) | 3(9.7%) | 7(53.8%) | |
| Partially Sufficient Data n:73 | 22(75.9%) | 7(63.6%) | 3(50%) | 8(80%) | 27(87.1%) | 6(46.2%) | |
| Completely Sufficient Data n:4 | 0(0%) | 0(0%) | 2(33.3%) | 1(10%) | 1(3.2%) | 0(0%) | |
| **DISCERN(Mean ± SD)** | 36.13 ± 14.22 | 32 ± 14.31 | 42 ± 18.5 | 47.40 ± 17.91 | 34.96 ± 13.69 | 33.07 ± 11.73 | 0.249 |
| Very Poor n:16 | 5(17.2%) | 4(36.4%) | 1(16.7%) | 0(0%) | 4(12.9%) | 2(15.4%) | |
| Poor n:55 | 15(51.7%) | 3(27.3%) | 2(33.3%) | 5(50%) | 21(67.7%) | 9(69.2%) | |
| Fair n:15 | 5(17.2%) | 4(36.4%) | 1(16.7%) | 1(10%) | 3(9.7%) | 1(7.7%) | |
| Good n:12 | 4(13.8%) | 0(0%) | 2(33.3%) | 3(30%) | 2(6.5%) | 1(7.7%) | |
| Excellent n:2 | 0(0%) | 0(0%) | 0(0%) | 1(10%) | 1(3.2%) | 0(0%) | |
| **HONcode** | | | | | | | 0.152 |
| + n:10 | 4(13.8%) | 0(0%) | 1(16.7%) | 7(70%) | 1(3.2%) | 1(7.7%) | |
| - n:90 | 25(86.2%) | 11(100%) | 5(83.3%) | 3(30%) | 30(96.8%) | 12(92.3%) | |
| **Reading Level** | | | | | | | 0.850 |
| Fairly easy to read | 0(0%) | 2(18.2%) | 0(0%) | 1(10%) | 3(9.7%) | 1(7.7%) | |
| Standart/Avarage n(%) | 4(13.8%) | 1(9.1%) | 2(33.3%) | 0(0%) | 2(6.5%) | 1(7.7%) | |
| Fairly difficult to read n(%) | 11(37.9%) | 4(36.4%) | 1(16.7%) | 3(30%) | 13(41.9%) | 5(38.5%) | |
| Difficult to read n(%) | 13(44.8%) | 4(36.4%) | 2(33.3%) | 6(60%) | 13(41.9%) | 6(46.2%) | |
| Very Diffucult to read n(%) | 1(3.4%) | 0(0%) | 1(16.7%) | 0(0%) | 0(0%) | 0(0%) | |
| **Readers Age** | | | | | | | 0.646 |
| 8–9 Years old (Fourth and Fifth Graders) n(%) | 0(0%) | 0(0%) | 0(0%) | 0(0%) | 1(3.2%) | 0(0%) | |
| 10–11 Years old (Fifth and Sixth graders) n(%) | 0(0%) | 0(0%) | 0(0%) | 0(0%) | 2(6.5%) | 0(0%) | |
| 11–13 Years old (Sixth and Seventh Graders) n(%) | 2(8.3%) | 1(9.1%) | 0(0%) | 1(10%) | 0(0%) | 0(0%) | |
| 12–14 Years old (Seventh and Eighth Graders) n(%) | 1(3.4%) | 2(18.2%) | 0(0%) | 0(0%) | 1(3.2%) | 1(7.7%) | |
| 13–15 Years old (Eighth and Ninth Graders) n(%) | 3(10.3%) | 1(9.1%) | 2(33.3%) | 1(10%) | 2(6.5%) | 2(15.4%) | |
| 14–15 Years old (Ninth to Tenth Graders) n(%) | 5(17.2%) | 2(18.2%) | 1(16.7%) | 1(10%) | 4(12.9%) | 4(30.8%) | |
| 15–17 Years old (Tenth to Eleventh Graders) n(%) | 8(27.6%) | 3(27.3%) | 0(0%) | 1(10%) | 10(32.3%) | 0(0%) | |
| 17–18 Years old (Twelfth Graders) n(%) | 11(37.9%) | 0(0%) | 0(0%) | 3(30%) | 6(19.4%) | 2(15.4%) | |
| 18–19 Years old (College Level Entry) n(%) | 0(0%) | 2(18.2%) | 1(16.7%) | 1(10%) | 1(3.2%) | 2(15.4%) | |
| 21–22 Years Old(college level) | 0(0%) | 0(0%) | 1(16.7%) | 1(10%) | 4(12.9%) | 0(0%) | |
| College Graduate n(%) | 1(3.4%) | 0(0%) | 1(16.7%) | 1(10%) | 0(0%) | 2(15.4%) | |

**Notes.**

JAMA, Journal of American Medical Association Benchmark Criteria; HONcode, The Health on the Net Foundation Code of Conduct (HONcode), Bold character; Statistically different ($p < 0.05$).

Ozduran and Büyükçoban (2022), *PeerJ*, DOI 10.7717/peerj.13686

**Table 4  Correlation relationships between rank and readability formulas, JAMA, DISCERN scores, HONcode precenses.**

| Rank | Alexa Rank | | Google Rank | | JAMA | | DISCERN | | GQS | | HONcode | |
|---|---|---|---|---|---|---|---|---|---|---|---|---|
| | r | p | r | p | r | p | r | p | r | p | r | p |
| Mean FRES | 0.178 | 0.084 | −0.007 | 0.946 | **−0.222** | **0.027** | **−0.223** | **0.026** | **−0.293** | **0.003** | −0.190 | 0.058 |
| Mean GFOG | −0.133 | 0.197 | 0.015 | 0.885 | **0.269** | **0.007** | **0.215** | **0.032** | **0.307** | **0.002** | **0.217** | **0.030** |
| Mean FKGL | **−0.211** | **0.039** | 0.007 | 0.945 | **0.226** | **0.024** | **0.200** | **0.046** | **0.275** | **0.006** | 0.185 | 0.066 |
| Mean CL Index | −0.134 | 0.191 | −0.043 | 0.670 | 0.166 | 0.099 | **0.205** | **0.041** | **0.261** | **0.009** | 0.133 | 0.189 |
| Mean SMOG index | −0.174 | 0.091 | 0.079 | 0.436 | **0.257** | **0.010** | **0.210** | **0.036** | **0.272** | **0.006** | 0.172 | 0.088 |
| Mean ARI | **−0.220** | **0.032** | −0.015 | 0.881 | **0.209** | **0.037** | 0.189 | 0.060 | **0.262** | **0.009** | 0.159 | 0.114 |
| Mean LW Formula | **−0.221** | **0.031** | 0.022 | 0.829 | **0.230** | **0.021** | 0.172 | 0.087 | **0.237** | **0.018** | 0.153 | 0.128 |
| Grade Level | −0.193 | 0.059 | 0.001 | 0.995 | **0.226** | **0.024** | **0.205** | **0.041** | **0.274** | **0.006** | −0.161 | 0.109 |
| JAMA | −0.032 | 0.100 | 0.088 | 0.385 | – | – | **0.670** | **0.001>** | **0.411** | **0.001>** | 0.131 | 0.194 |
| DISCERN | −0.028 | 0.784 | −0.104 | 0.302 | **0.670** | **0.001>** | – | – | **0.765** | **0.001>** | **0.287** | **0.004** |
| GQS | −0.063 | 0.539 | −0.222 | 0.027 | **0.411** | **0.001>** | **0.765** | **0.001>** | – | – | **0.362** | **0.001>** |
| HONcode | −0.114 | 0.268 | −0.076 | 0.451 | 0.131 | 0.194 | **0.287** | **0.004** | **0.362** | **0.001>** | – | – |

**Notes.**

FRES, Flesch reading ease score; FKGL, Flesch-Kincaid grade level; SMOG, Simple Measure of Gobbledygook; GFOG, Gunning FOG; CL, Coleman-Liau score; ARI, automated readability index; LW, ve Linsear Write; HONcode, The Health on the Net Foundation Code of Conduct (HONcode); JAMA, Journal of American Medical Association Benchmark Criteria.

Bold character; statistically different ($p < 0.05$).

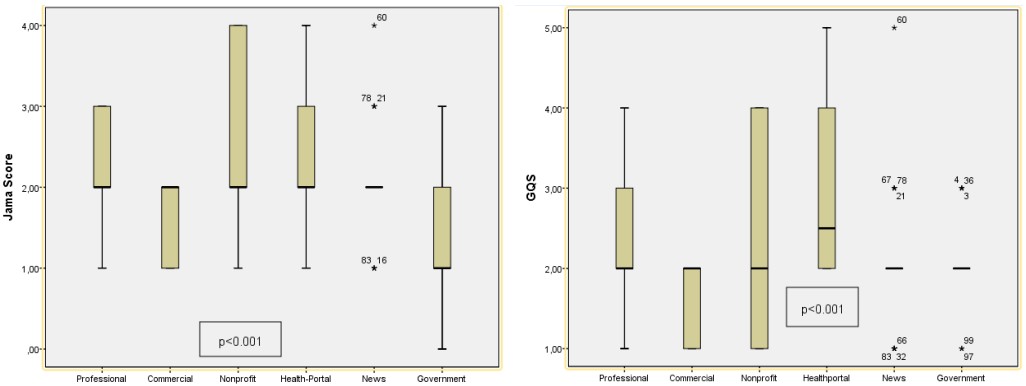

**Figure 3 Evaluation of JAMA reliability and GQS quality scores of websites according to their typology.** The *P* value indicates whether there is a significant difference in readability according to typologies (*p* < 0.05).

organization, Health Portal, News, Professional, Commercial, Government. According to the website typologies, the quality (GQS) ranking is as follows, from the highest to the lowest; Health Portal, Non-Profit organization, Professional, Government, News, Commercial (Fig. 3).

## Readability evaluation

In the analysis of the text contents readability of the 100 websites, the mean FRES was 51.40 ± 10.65 (difficult), mean GFOG was 13.14 ± 2.16 (very difficult), mean FKGL and SMOG were 10.93 ± 2.17 and 9.83 ± 1.66 years of education, respectively, mean CL index was 10.62 ± 1.71 years of education, and mean ARI index was 11.03 ± 257 years of education. According to the FRES, GFOG and Coleman results, $n = 17(17\%)$ websites scores were found to be above 60 points and their readabilities are below the sixth grade level. According to Linsear readability results, 3(3%) websites, according to FKGL and ARI scores 5(5%) websites, according to SMOG results 6(6%) websites are at sixth grade level and below. There was no significant relationship in a comparison of the website type and all the readability indices (FRES, $p = 0.669$; GFOG, $p = 0.520$; FKGL, $p = 0.467$; CL, $p = 0.860$; SMOG, $p = 0.447$; ARI, $p = 0.517$) (Fig. 4). There was a significant difference upon comparison of the mean readability index scores of the 100 websites and the sixth-grade reading level ($p < 0.001$) (Table 3). There was no significant difference between the reading level and readers age of the websites by type (Reading Level $p = 0,850$; Readers Age $p = 0,646$).

## Correlation analysis

There was a weak positive correlation between the mean readability scores based on the readability formulas and JAMA reliability scores, DISCERN quality scores, and GQS values (Table 5). There was a weak positive correlation between the JAMA and DISCERN scores ($r = 0.670$, $p < 0.001$) and GQS scores ($r = 0.411$, $p < 0.001$).

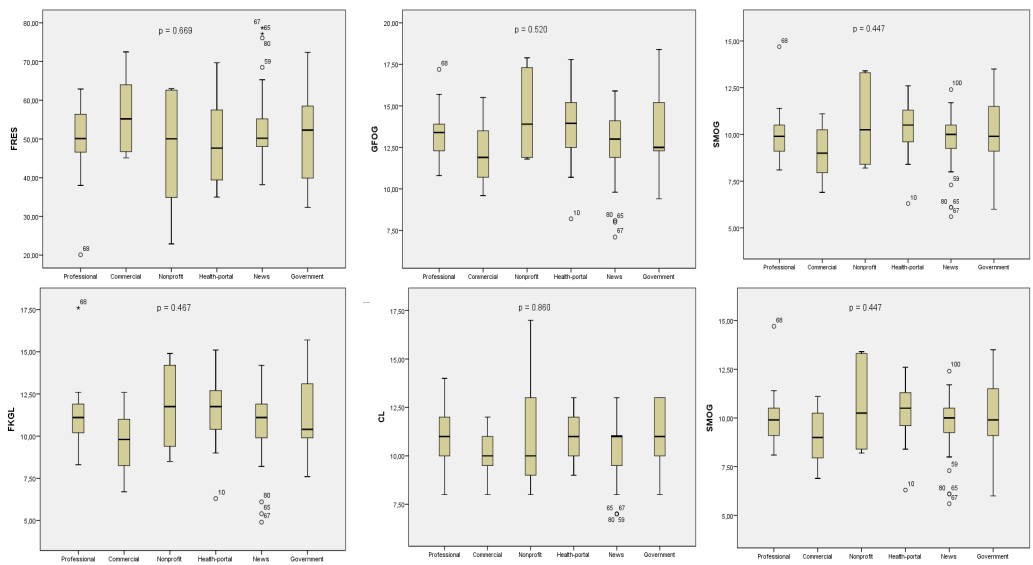

**Figure 4 Evaluation of JAMA reliability and GQS quality scores of websites according to their typology.** The *P* value indicates whether there is a significant difference in quality and reliability according to typologies (*p* < 0.05).

## Popularity ranking

The mean Alexa ranking of the 100 websites was 287,786.94 ± 798,542.83, respectively. There was a statistically significant difference between ALEXA values and website types, and it was determined that this significant difference was related to commercial websites (*p* < 0.001).

## Website contents

According to the content analysis, the numbers of topics included by the websites were as follows; $n = 29(29\%)$ etiology, $n = 28(28\%)$ diagnosis, $n = 85(85\%)$ non-pain symptoms, $n = 60(60\%)$ treatment, $n = 29(29\%)$ exercise, $n = 16(16\%)$ prevention, $n = 20(20\%)$ risk factors, and $n = 17(17\%)$ vaccine pain relationship. There was no significant difference between the top 10 and the remaining websites upon content analysis (Etiology, $p = 0,160$; Diagnosis, $p = 0,981$; Non-pain-related symptoms, $p = 0,059$; Treatment, $p = 0,572$; Exercise, $p = 0,081$, Prevention, $p = 0,149$; Risk factors, $p = 0,168$; Vaccine–pain relationship, $p = 0,186$) (Table 6).

## DISCUSSION

This study investigated whether Internet-based PEMs on pain after COVID-19 infection were reliable, of high quality, and readable. Furthermore, it also investigated which website types provided highly reliable and readable information. Accordingly, a comparison of the 10 most visited sites on the first page with the remaining websites that appeared on the search engine based on quality, reliability, and readability ratings was conducted. Finally,

Ozduran and Büyükçoban (2022), *PeerJ*, DOI 10.7717/peerj.13686

**Table 5** All group of websites' mean results and statistical comparison of text content to 6th grade reading level.

| Readability Indexes | Top 10 (n = 10) Mean ± SD | Others (n = 90) Mean ± SD | Total (n = 100) Mean ± SD | Comparison of the first 10 websites and remaining 90 websites according to parameters (p) | Comparison of the 100 websites' according to 6th grade reading level(p) |
|---|---|---|---|---|---|
| FRES | 55.73 ± 10.13 | 50.92 ± 10.65 | 51.40 ± 10.65 | 0.125 | **<0.001** |
| GFOG | 12.58 ± 2.32 | 13.20 ± 2.15 | 13.14 ± 2.16 | 0.601 | **<0.001** |
| FKGL | 10.22 ± 2.24 | 11.01 ± 2.16 | 10.93 ± 2.17 | 0.202 | **<0.001** |
| The CL Index | 10.10 ± 1.72 | 10.67 ± 1.71 | 10.62 ± 1.71 | 0.274 | **<0.001** |
| The SMOG Index | 9.09 ± 1.74 | 9.91 ± 1.64 | 9.83 ± 1.66 | 0.190 | **<0.001** |
| ARI | 10.46 ± 2.69 | 11.09 ± 2.56 | 11.03 ± 2.57 | 0.334 | **<0.001** |
| LW Formula | 11.78 ± 3.19 | 12.59 ± 2.87 | 12.51 ± 2.90 | 0.569 | **<0.001** |
| Grade Level | 10.40 ± 2.17 | 11.05 ± 2.09 | 10.99 ± 2.10 | 0.279 | **<0.001** |
| **Popularity Index** | | | | | |
| Alexa Rank | 48,387.77 ± 56,870.73 | 312,552.37 ± 835,155.40 | 287,786.94 ± 798,542.83 | 0.806 | |
| **JAMA Mean ± SD** | 1.80 ± 1.03 | 2.02 ± 0.73 | 2 ± 0.76 | 0.350 | |
| **DISCERN Mean ± SD** | 38.20 ± 10.68 | 36.20 ± 15.11 | 36.40 ± 14.70 | 0.498 | |
| **GQS Mean ± SD** | 2.70 ± 0.48 | 2.12 ± 0.87 | 2.18 ± 0.85 | **<0.001** | |
| **JAMA** | **n(%)** | **n(%)** | | | |
| Insufficient Data | 4(40%) | 19(21.1%) | 23(23%) | *0.350 | |
| Partially Sufficient Data | 6(60%) | 67(74.4%) | 73(73%) | | |
| Completely Sufficient Data | 0(0%) | 4(4.4%) | 4(4%) | | |
| **DISCERN** | **n(%)** | **n(%)** | **n(%)** | | |
| Very Poor n(%) | 0(0%) | 16(17.8%) | 16(16%) | | |
| Poor n(%) | 7(70%) | 48(53.3%) | 55(55%) | 0.498 | |
| Fair n(%) | 2(20%) | 13(14.4%) | 15(15%) | | |
| Good n(%) | 1(10%) | 11(12.2%) | 12(12%) | | |
| Excellent n(%) | 0(0%) | 2(2.2%) | 2(2%) | | |
| **HONcode n(%)** + | 2(20%) | 8(8.9%) | 10(10%) | 0.262 | |
| − | 8(80%) | 82(91.1%) | 90(90%) | | |
| **GQS** | **n(%)** | **n(%)** | **n(%)** | | |
| Low Quality | 3(30%) | 72(80%) | 75(75%) | **<0.001** | |
| Medium Quality | 7(70%) | 10(11.1%) | 17(17%) | | |
| High Quality | 0(0%) | 8(8.9%) | 8(8%) | | |

Ozduran and Büyükçoban (2022), *PeerJ*, DOI 10.7717/peerj.13686

**Table 5** (*continued*)

| Readability Indexes | Top 10 ($n = 10$) | Others ($n = 90$) | Total ($n = 100$) | Comparison of the first 10 websites and remaining 90 websites according to parameters (p) | Comparison of the 100 websites' according to 6th grade reading level(p) |
|---|---|---|---|---|---|
| | Mean ± SD | Mean ± SD | Mean ± SD | | |
| **Typology** | **n(%)** | **n(%)** | **n(%)** | | |
| Professional | 6(60%) | 23(25.6%) | 29(29%) | | |
| Commercial | 0(0%) | 11(12.2%) | 11(11%) | | |
| Non-profit | 0(0%) | 6(6.7%) | 6(6%) | **0.043** | |
| Health portal | 1(10%) | 9(10%) | 10(10%) | | |
| News | 0(0%) | 31(34.4%) | 31(31%) | | |
| Government | 3(30%) | 10(11.1%) | 13(13%) | | |

**Notes.**

FRES, Flesch reading ease score; FKGL, Flesch-Kincaid grade level; SMOG, Simple Measure of Gobbledygook; GFOG, Gunning FOG; CL, Coleman-Liau score; ARI, automated readability index; LW, ve Linsear Write; JAMA, Journal of American Medical Association Benchmark Criteria; HONcode, The Health on the Net Foundation Code of Conduct (HONcode); GQS, Global Quality Score.
Bold character; statistically different ($p < 0.05$).

**Table 6 Content analysis by typology.**

| | | Professional | Commercial | Non-Profit | Health Portal | News | Government | p |
|---|---|---|---|---|---|---|---|---|
| Etiology | + | 8(27.6%) | 1(9.1%) | 3(50%) | 1(10%) | 13(41.9%) | 3(23.1%) | 0.160 |
| | − | 21(72.4%) | 10(90.9%) | 3(50%) | 9(90%) | 18(58.1%) | 10(76.9%) | |
| Diagnosis | + | 9(31%) | 3(27.3%) | 2(33.3%) | 3(30%) | 7(22.6%) | 4(30.8%) | 0.981 |
| | − | 20(69%) | 8(72.7%) | 4(66.7%) | 7(70%) | 24(77.4%) | 9(69.2%) | |
| Non-pain symptoms | + | 28(96.6%) | 8(72.7%) | 5(83.3%) | 6(60%) | 27(87.1%) | 11(84.6%) | 0.059 |
| | − | 1(3.4%) | 2(18.2%) | 1(16.7%) | 4(40%) | 4(12.9%) | 1(7.7%) | |
| Treatment | + | 17(58.6%) | 9(81.8%) | 4(66.7%) | 7(70%) | 16(51.6%) | 7(53.8%) | 0.572 |
| | − | 12(41.4%) | 2(18.2%) | 2(3.3%) | 3(30%) | 15(48.4%) | 6(46.2%) | |
| Exercise | + | 7(24.1%) | 6(54.5%) | 3(50%) | 5(50%) | 6(19.4%) | 2(15.4%) | 0.081 |
| | − | 22(75.9%) | 5(45.5%) | 3(50%) | 5(50%) | 25(80.6%) | 11(84.6%) | |
| Prevention | + | 3(10.3%) | 1(9.1%) | 1(16.7%) | 10(100%) | 6(19.4%) | 5(38.5%) | 0.149 |
| | − | 26(89.7%) | 10(90.9%) | 5(83.3%) | 0(0%) | 25(80.6%) | 8(61.5%) | |
| Risk Factors | + | 6(20.7%) | 0(0%) | 2(33.3%) | 3(30%) | 4(12.9%) | 5(38.5%) | 0.168 |
| | − | 23(79.3%) | 11(100%) | 4(66.7%) | 7(70%) | 27(87.1%) | 8(61.5%) | |
| Vaccine-pain relationship | + | 6(20.7%) | 0(0%) | 1(16.7%) | 1(10%) | 4(12.9%) | 5(38.5%) | 0.186 |
| | − | 23(79.3%) | 11(100%) | 5(83.3%) | 9(90%) | 27(87.1%) | 8(61.5%) | |

**Notes.**

Statistically different($p < 0.05$).

the relationship between the websites' readability and the quality and reliability thereof was assessed.

Pain is one of the important symptoms associated with COVID-19. Widespread organ and tissue damage, especially in the musculoskeletal system, and increased cytokine levels due to infection, have been suggested with regard to the etiology and pathogenesis of pain (*Su et al., 2020*). A meta-analysis suggested myalgia and headache as the most prevalent musculoskeletal and neurological symptoms, respectively (*Abdullahi et al., 2020*). In the case of long COVID-19, a term used for patients with symptoms persisting more than 1 month, it was noted that pain symptoms were among the other persistent symptoms that lasted for a prolonged duration. It is reported that the pain symptoms associated with COVID-19 persisted for 1–11 weeks (*Şahin et al., 2021*). *Tetik et al. (2021)* found that the rate of post-COVID-19 pain was 7.9%, and it was mostly observed in patients with advanced age and often with long-term clinical manifestations of body, lumbar, and joint pain and headache.

Written communication has been demonstrated to be an indispensable tool in times of crisis. It is imperative to ensure the comprehensibility of the emergency messages during such times. Relevant studies suggested that written messages could be more readily and accurately remembered than verbal messages (*Edworthy et al., 2015*). Whether they are verbal or written, accurate and reliable messages should be quickly disseminated across the society and easily understood by the majority (*Edworthy et al., 2015*). It was suggested that for healthcare-related information to be most effective for the general public, such information must be aligned to the sixth-grade readability level (*Basch et al., 2020*). Text

contents comprised of long and complex sentences may impair the reader's self-confidence while trying to obtain medical information and cause them to give up reading the text. The National Literacy Institute of the US Department of Education reported that 32 million US adults were illiterate and 68 million Americans had a reading level below the fifth grade (*Daraz et al., 2018*). Considering that the acquisition of Internet-based healthcare-related information has increased, providing more readable information on websites will help individuals protect against diseases and quickly assess diagnosis and treatment processes when they are ill. *Basch et al. (2020)* suggested that the readability level of Internet-based information on COVID-19 was much higher and more difficult compared with that of an average American citizen.

In the present study, there was a significant difference between the top 10 and the remaining websites in a comparison of the websites based on their type. Websites that were classified in the news and professional types most frequently appeared in the search results. Nevertheless, websites created by professional institutions constituted the majority of the top 10 searches that appeared on the first page on the Google search engine. A significant difference was found between the website types and reliability scores. It was concluded that this difference was associated with higher JAMA scores in the websites created by nonprofit organizations. There was also no significant difference in reliability between the top 10 and the remaining websites. Nevertheless, there was a significant difference in quality based on the GQS values between the top 10 and the remaining websites. It was noted that 70% of the top 10 websites were of medium quality, whereas 72% of the remaining 90 sites were of low quality. There was no significant difference between the websites by type in an assessment of readability indices. There was no significant difference between the top 10 and the remaining websites in terms of readability indices.

The majority of the websites included in the present study were created by news channels. In a study of online information about COVID-19, *Kłak et al. (2022)* reported that news websites constituted the largest group, similar to the results of the present study. All COVID-19-related developments, daily case and death figures, and vaccination statistics were shared instantly by news channels and news websites during the pandemic. The audience statistics suggest that news websites also maintain their ranking with regard to the topic of post-COVID-19 pain. Content on COVID-19-related epidemiology and isolation are frequently shared on news websites. This raises social concerns and lack of trust. People try to remedy their lack of trust by accessing information via the Internet. Governments have taken steps to compensate for the lack of Internet-based information, and accordingly, government-affiliated websites were introduced (*e.g.*, Robert Koch Institute in Germany). A study by *Okan et al. (2020)* spanning from March to April 2020 reported that the information made available by local authorities prevented information pollution .

In the present study, 10 of the 100 websites were HONcode-compliant. *Valizadeh-Haghi, Khazaal & Rahmatizadeh (2021)* investigated the credibility of health websites on COVID-19 and found that 12.8% of the websites included in their study were HONcode-compliant. The results of the present study are consistent with those reported in the relevant literature. HONcode is the earliest and most frequently used code of ethics and reliability intended for medical and healthcare-related information available on the Internet (*Valizadeh-Haghi,*

*Khazaal & Rahmatizadeh, 2021*). Accordingly, the HONcode-compliant websites had higher DISCERN and JAMA scores in the present study. An implication of the above is that healthcare professionals may advise their patients to prefer HONcode-compliant websites when seeking Internet-based information about post-COVID-19 pain.

In the present study, the overall mean DISCERN score of the websites was considered "poor" (36.40 ± 14.70). Similar to the present study, *Halboub et al. (2021)* reported the same score as 31.5 ± 12.55 in a study on healthcare information related to COVID-19. The fact that certain web sources, including academic or scientific journals, were not excluded in such studies in the relevant literature as the study by *Kłak et al. (2022)* which reported high DISCERN scores, may result in higher DISCERN scores as well as high in readability scores. It is well established that patients prefer sources with less medical terminology and better readability when they need to access Internet-based healthcare-related information. Whereas, academic resources are intended for use among the healthcare professionals and aim to make a scientific contribution.

There was no significant difference in a comparison of the website types and readability. The average readability results were found to be well above the sixth-grade reading level recommended by the National Institutes of Health *Kłak et al. (2022)*. *Jayasinghe et al. (2020)* who excluded academic websites in their investigation of the quality and readability of online information about COVID-19, reported moderate-to-low readability scores. Ensuring easier readability levels may help with reaching wider audiences, and the power of information can be presented more effectively based on an appropriate readability level matching that of the general public.

An assessment based on the content indicated that most of the websites (85%) included information about non-pain symptoms, followed by 60% of the websites with information on treatment. There was no significant difference between the website types and topics. The most frequent topics were pulmonary symptoms, followed by social distancing in the relevant literature, which investigated online information about COVID-19. Pain and other non-respiratory symptoms were ranked fifth in their studies (*Jayasinghe et al., 2020*). Considering that the topics of prevention, treatment and vaccination were alternated during the COVID-19 pandemic, up-to-date popular topics were reflected on the websites during each period and presented to the attention of visitors.

## Limitations of this study

There are limitations to this study. These limitations include the search of websites in English language, use of a single search engine, and inclusion of websites that use the data network of a single country. There is no consensus on the gold standard readability index in the assessment of the readability of Internet-based patient education materials; nevertheless, the indices used in this study were among the most frequently used formulas, which, in the present study, indicated that the websites were intended for an educational level far above the recommended level. The readability, reliability and quality of the websites were evaluated over precise scales and criteria, and the same results were obtained among the authors at a rate of 98%. Although there is a 2% difference, there may be a bias between the authors, which is another limitation of our study.

### Strengths of this study

In our study, we examined an important topic about the ongoing pain of those who had Covid-19 infection despite the end of the infection. In this period when people stay at home and try to get information over the internet, we tried to determine whether the information on the internet is accurate, high quality and reliable. During our study, we tried to evaluate the websites that are used more by the public by excluding academic websites.

## CONCLUSION

The readability level of Internet-based PEMs on post-COVID-19 pain was considerably higher than the sixth-grade level recommended by the National Institutes of Health. The website contents had low reliability and poor quality. The websites of nonprofit organizations provided more reliable information, the health portals offered information of higher quality, and the news websites ranked lowest in all the parameters. The correlation between JAMA and DISCERN scores and HONcode compliance suggested that reliable websites also provided high-quality information. During the development of healthcare-related websites intended for the general public on the COVID-19 pandemic in the first quarter of the 21st century, the language of the website should be checked against the relevant readability indices, the website should maintain a readability level that fits the average education level of the relevant country or countries that are the intended recipient of the information, and the website should contain high-quality and reliable information. Authorities dealing with health and drug informatics have a great responsibility to present reliable, quality and readable information for the public while preparing their websites.

### Funding

The authors received no funding for this work.

### Competing Interests

The authors declare there are no competing interests.

### Author Contributions

- Erkan Ozduran conceived and designed the experiments, performed the experiments, analyzed the data, prepared figures and/or tables, authored or reviewed drafts of the article, and approved the final draft.
- Sibel Büyükçoban conceived and designed the experiments, analyzed the data, prepared figures and/or tables, authored or reviewed drafts of the article, and approved the final draft.

### Ethics

The following information was supplied relating to ethical approvals (i.e., approving body and any reference numbers):

The University of Dokuz Eylul granted Ethical approval to carry out the study within its facilities (Ethical Application Ref: GOA 2022/06-08).

## Data Availability

The raw measurements are available in the Supplementary File.

## Supplemental Information

Supplemental information for this article can be found online at http://dx.doi.org/10.7717/peerj.13686#supplemental-information.

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

Indianapolis: New Riders.