# Peer review of "Evaluating the readability, quality and reliability of online patient education materials on post-covid pain"

_PeerJ, doi:10.7717/peerj.13686_

## Round 0.1 · original submission · Major Revisions

The reviewers have done a thorough job and I particularly draw your attention to Reviewer 1's points because they are in my view the most significant issues with regard to the robustness of your results.

·

Basic reporting

There is some unconventional language used, and although you can understand the point that the authors are trying to make, it is made in a slightly unorthodox way. I made some specific comments below.

Some specific comments.
Introduction
Line 57; late 2019 was when it was first detected, the medical and humanitarian crisis was later. It is emotive language, it would be really great to accompany such language with some numbers, for example, percentage of the population affected after 3 months or the number of deaths in the first month.

Line 71- It is evident that the correct treatment algorithm would help individuals with recovery given that pain symptoms adversely affect the quality of life during the post-COVID period.
This sentence seems a bit out of context, could you elaborate or clarify your point about pain symptoms. These symptoms are not mentioned earlier in the paragraph. I think this paragraph should focus more on post covid pain as that is a main aim of the manuscript that seems to get a bit lost.

Line 74; Patients can rapidly access the desired healthcare content using Internet-based patient educational materials (PEM), which have recently been used as an important tool for acquiring further information.
Could you add some references to this sentence?

Line 80 typo
Levelx

Line 80; In case the readability of online information posted on a website is above the said grade, it may be considered difficult to read and understand for an average reader. Therefore, it is important that
Suggest change to, “if the readability

Line 88 A study by Worrall et al. reported that the readability level of online information about COVID-19 90 was poor and difficult to read (Worrall et al., 2020).
Could you be more specific here? What do you mean by poor and difficult to read? Could you provide numbers?

Line 91; It is well established that patients furnished with information about the etiology, 92 pathophysiology, treatment, and prevention methods would more likely participate in and 93 comply with the disease prevention or treatment procedures.
Is it? Could you provide a reference here.

Line 93; It is evident that providing 94 individuals with reliable, high-quality, and readable online information about post-COVID-19 95 pain would help with the management of a condition that affects many people.
Is it? Again this needs a reference

Methods
Line 116; Could you clarify what you mean by a collective assessment?

Line 118; What sector?

Line 139; Could you provide some more information on how it was decided to categorise the websites?

Line 191; Could you restate the aforementioned exclusions

Line 201; Content analyses

Line 208; For statistical analysis, the Mann–Whitney U or Kruskal 209 Wallis tests were used to compare groups with continuous values such as readability indices and 210 sixth class level. For comparison of frequency variables, the Chi-square or Fisher exact tests 211 were used. A p-value lower than 0.05 was accepted as a statistically significant difference.
I don’t understand the value of these statistical tests. It seems that they are to provide more weight/ impact or meaning to the study, but it seems a bit like manufacturing importance to me. I wonder if they are necessary, would it be just as informative to report numbers and percentages. Aren’t you just using different measures of readability, I don’t understand the value of knowing if there was a difference in the readability scores, isn’t it more important what the readability score was. Is there a gold standard readability score, perhaps it is best just to choose that and report the average readability score for the websites included?

Results
The authors do not seem to report on the content analysis outlined in the methods

I would expect the results to follow the same format as they are outlined in the methods.

Discussion
Line 285 this paragraph would be more useful in the introduction, as the information and rationale for investigating post-Covid pain information is not well-established in the introduction as it is.

Experimental design

The idea behind the design is good but I think the authors attempt to cover too much and in doing so make the analysis unnecessarily complicated. For example, they measure readability using 7 independent readability indices. They also add some statistical analysis of questionable value. I wonder if it would be just as informative to report numbers and percentages. For example, the authors report whether or not there is a statistical difference in the readability scores from the different indices. I don’t understand the value of knowing if there was a difference in the readability scores, that seems to be assessing the validity of the tool to assess readability rather than readability itself, isn’t it more important what the readability score was. Is there a gold standard readability score, perhaps it is best just to choose that and report the average readability score for the websites included?
There is insufficient detail on some aspects of how the analyses are conducted, for example, the authors report conducting a content analysis but with minimal information on how that was conducted.

Validity of the findings

There seems to be some value in the findings, but it is buried in a over-complicated analyses of association and statistical significance. I think the paper could be quite interesting if the authors restricted their analysis to simple measures of numbers and percentages, and illustrated their findings with box plots.

Additional comments

There is no doubt that the authors have worked hard on this paper but I think it would require significant revision before being accepted.

Reviewer 2 ·

Basic reporting

Thanks for inviting to review this master piece
This manuscript is well presented but lacks scientific writing at some points.
Suggesting proof reading, add literature references, improve rationale of study by stating research gap
Write COVID-19 instead of COVID throughout the manuscript

Experimental design

This study is well-designed but, there are a few things that should be considered in the method section.
How sample size for websites calculated??? kindly mention in manuscript
add more information about the study type in the method section.
Add heading on ethical considerations
heading on websites selection criteria
in Content Analyse, The websites were investigated and assessed , WHICH TYPE OF WEBSITES?? MENTION NAME OF WEBSITES AND ADD REFERENCES
Please write dependent and independent variables in the analysis section

Validity of the findings

The results of this study are well presented but author need to make headings of results like readability, quality, reliability, and sub-headings in an adequate manner.
If possible make bar chart of Figure 1
The discussion section is well explained

Add limitations of this study under separate heading.
Add strengths of this study under separate heading.

Additional comments

None

·

Basic reporting

Thanks for the invitation to review this study. This is an interesting piece of work where the authors tried to estimate the validity and reliability of PEM which was used by the general population to address the post-covid pain. There are a few concerns that should be addressed before making any decision on this submission.
1. Please provide the rationale why post-covid-pain was selected as a topic of search or discussion, as the population was also more interested in seeking information on the treatment (either allopathic or herbal) of COVID-19.

Experimental design

Authors have used various criteria or tools to assess the reliability, popularity, readability, and quality of the online websites used to seek information on post-covid pain. It would be better and reader-friendly if the authors could provide a table with the name of these tools along with the parameters these tools are evaluating.
Why did the authors just limit the search to 100 websites? Google also provides results according to the region where the search is being performed. How the authors will exclude this bias to make the findings more generalizable.
Please provide the study flow diagram for a more detailed and comprehensive understanding of the study methodology.

Validity of the findings

The authors have presented the mean values with commas. The mean values and standard deviations are usually presented with a full stop for decimal values. Please consider the amendments.
I did not find that authors have performed the search and evaluation of the websites in duplicates or triplicates (two or more authors perform the process independently). If the search and evaluation are not performed in duplicates then how the researcher's bias can be omitted from the findings. There is a need for clarification.
Can authors provide the rank of reliability and quality of five groups presented in the current study in order to see which sources are more reliable and were of quality for healthcare-related information? This data is presented but not in a ranked way.

Additional comments

The limitation section should also discuss the involvement of bias as described in the above comments. Authors need to provide suggestions to authorities dealing with health and drug informatics.

---

## Round 0.2 · accepted · Accept

Congratulations on this paper and the extensive changes you have made in response to what were excellent reviews.

Reviewer 2 ·

Basic reporting

No Comments, well revised, accept in current form

Experimental design

No Comments, well revised, accept in current form

Validity of the findings

No Comments, well revised, accept in current form

·

Basic reporting

The authors have addressed my comments. I have no more concerns.

Experimental design

The authors have addressed my comments. I have no more concerns.

Validity of the findings

The authors have addressed my comments. I have no more concerns.

Additional comments

The authors have addressed my comments. I have no more concerns.